# Gut Microbiota and Critical Metabolites: Potential Target in Preventing Gestational Diabetes Mellitus?

**DOI:** 10.3390/microorganisms11071725

**Published:** 2023-06-30

**Authors:** Runan Hu, Zhuo Liu, Yuli Geng, Yanjing Huang, Fan Li, Haoxu Dong, Wenwen Ma, Kunkun Song, Mingmin Zhang, Yufan Song

**Affiliations:** 1Institute of Integrated Traditional Chinese and Western Medicine, Tongji Hospital, Tongji Medical College, Huazhong University of Science and Technology, Wuhan 430030, China; hurunan19970109@163.com (R.H.); zhuoliutjzx@126.com (Z.L.); yuligeng@hust.edu.cn (Y.G.); hyj13260684620@163.com (Y.H.); d202182041@hust.edu.cn (F.L.); 2Department of Integrated Traditional Chinese and Western Medicine, Tongji Hospital, Tongji Medical College, Huazhong University of Science and Technology, Wuhan 430030, China; donghx4315@yeah.net (H.D.); wwmahusttj@163.com (W.M.); songkunkun@tjh.tjmu.edu.cn (K.S.); mmzhangeins@163.com (M.Z.)

**Keywords:** gestational diabetes mellitus, gut microecology, gut microbiota, short-chain fatty acids, interaction

## Abstract

Gestational diabetes mellitus (GDM) is an intractable issue that negatively impacts the quality of pregnancy. The incidence of GDM is on the rise, becoming a major health burden for both mothers and children. However, the specific etiology and pathophysiology of GDM remain unknown. Recently, the importance of gut microbiota and related metabolic molecules has gained prominence. Studies have indicated that women with GDM have significantly distinct gut microbiota and gut metabolites than healthy pregnant women. Given that the metabolic pathways of gut flora and related metabolites have a substantial impact on inflammation, insulin signaling, glucose, and lipid metabolism, and so on, gut microbiota or its metabolites, such as short-chain fatty acids, may play a significant role in both pathogenesis and progression of GDM. Whereas the role of intestinal flora during pregnancy is still in its infancy, this review aims to summarize the effects and mechanisms of gut microbiota and related metabolic molecules involved in GDM, thus providing potential intervention targets.

## 1. Introduction

Gestational diabetes mellitus (GDM) is the most common metabolic complication of pregnancy, affecting up to 25.5% of pregnant women [1,2]. It is defined as glucose intolerance provoking a transient state of hyperglycemia early or first detected during pregnancy [3]. GDM greatly increases the odds of polyhydramnios, pre-eclampsia, macrosomia, shoulder dystocia, cesarean section, and neonatal death [1,4,5,6,7]. Even with a successful birth, children may have a higher risk of developing obesity, diabetes, cardiometabolic disorders, and other metabolic disorders in the future [1,3]. Evidence suggests that women diagnosed with GDM in their first pregnancy have a 34.5% increased risk of developing GDM in subsequent pregnancies when compared to healthy women [8], and their prevalence of T2DM diagnosed within 5 years is approximately 20–50% [9,10,11]. Therefore, GDM has become a major health burden for both mothers and children.

Until now, the specific etiology and pathophysiology of GDM are still unknown. Profound hormonal, metabolic, and immunological changes that occur during pregnancy may be important contributors to GDM [7]. The first trimester is dominated by anabolism, storing adequate energy for fetal development. At this time, maternal insulin secretion elevates, glucose uptake by adipose tissue rises, and pregnant women begin to gain weight. As the pregnancy progresses, levels of placental and metabolic hormones, as well as proinflammatory cytokines, tend to increase, reducing maternal insulin sensitivity in the second half of pregnancy. Maternal insulin insensitivity stimulates gluconeogenesis and lipolysis during the third trimester, which is dominated by catabolism, resulting in elevated levels of maternal plasma glucose and free fatty acids (FFA) that are transported across the placenta and provide adequate energy for fetal development. However, hyperglycemia occurs during this process in susceptible pregnant women who are unable to compensate for insulin resistance (IR) and insufficient insulin secretion, thus resulting in GDM [3,4,7,12].

The 16S sequencing method, metabolomics, and next-generation sequencing (NGS) platforms all assist us in better comprehending the relationship between gut microecology and diseases [7]. The same holds true for GDM. Gut microecology means the great diversity of microorganisms and their surrounding environment. Gut microbiota is the most important component of gut microecology, although intestinal mucosa has a significant impact on gut microecology [13]. Through digestion, vitamin synthesis, metabolism, and systemic immunomodulation, the gut microbiota, also known as the “second genome,” play a crucial role in maintaining host physiology and homeostasis [7,14,15,16]. The gut metabolites, such as bile acids (BAs), amino acids, and short-chain fatty acids (SCFAs), are key mediators in host-microbiota cross-talk, modulating the integrity of the gut barrier, host physiology, and host pathological changes [16]. Studies have indicated that women with GDM have significantly distinct gut microbiota and gut metabolites than healthy pregnant women [9,17,18]. Moreover, some changes in maternal gut flora in GDM patients may be transmitted vertically to their children [19]. Given that gut flora and related metabolites substantially impact inflammation, insulin signaling, and glucose and lipid metabolism [15,16,20], GDM might be connected to disruptions in the gut microbiota and related metabolites, especially SCFAs. The purpose of this review is to summarize the impacts and mechanisms of gut microbiota and related metabolites in GDM, as well as to suggest prospective intervention targets, thereby expanding the chances for GDM prevention.

## 2. Gut Microbiota Changes during Normal Pregnancy

Firmicutes and Bacteroidetes are the most abundant forms of bacteria in gut flora, followed by Actinomycetes and Proteobacteria [21]. To prepare for the newborn, profound hormonal, immunological, and metabolic modifications occur throughout pregnancy [22,23], affecting the immunological state and energy storage of the intestinal mucosa [24,25], thus altering the composition of the gut microbiome and their metabolites [25,26,27].

Most studies agreed that the abundance of gut bacteria associated with energy storage promotion and inflammatory states might increase during pregnancy. Butyrate-producing bacteria such as *Faecalibacterium* and *Eubacterium* were identified to be over-represented in the first trimester [24]. At the same time, bacteria related to energy storage promotion, such as *Akkermansia*, *Bifidobacterium*, and Firmicutes, were hypothesized to be increased in the third trimester, and proinflammatory bacteria Proteobacteria and Actinobacteria were also augmented [25,28,29]. Santacruz et al. concurred that the abundance of gut bacteria associated with inflammatory states was increased in pregnancy [30]. Moreover, bacterial richness (α-diversity) was eventually diminished toward the third trimester [7,25]. Fecal samples from the first and third trimesters were pooled from five healthy-weight pregnant women, which were subsequently transplanted into the germ-free wild-type Swiss-Webster female mice. Shotgun metagenomics analysis revealed that the fraction of bacteria involved with evoking inflammatory states and energy storage was elevated during pregnancy. However, Digiulio et al. concluded minimal remodeling of gut microbiome associated with pregnancy [31] and intestinal flora in the first trimester was equivalent to that of non-pregnant women [30,32].

## 3. Gut Microbiota and Related Metabolic Changes during GDM

Pregnant women with GDM are characterized by higher IR and glucose intolerance during pregnancy as compared to healthy pregnant women [33]. Previous studies have shown that the gut microbiome can regulate metabolic homeostasis, affecting the composition of the gut microbiome [9,34]. Therefore, it is plausible to believe that the changed composition of intestinal flora and related metabolites in GDM patients may be the underlying mechanism of GDM.

Emerging studies have concentrated on alterations of intestinal flora and related metabolites during GDM. Most studies have revealed that women with GDM had significant differences in beta diversity (the overall structure of the microbiota) and lower alpha diversity (the number of species present in the given community) [17,18,35,36,37,38,39]. However, no difference in beta diversity and alpha diversity or an increase in alpha diversity across the pregnant trimesters was also reported [40,41,42,43].

Changes appear in early pregnancy. A case-control study with 75 overweight participants that thoroughly removed any confounding factors discovered that the relative abundance of the *Ruminococcaceae* family was significantly augmented during early pregnancy in women who later progressed to GDM compared to those without developing GDM [44]. Greater *Ruminococcaceae* family abundance was related to impaired glucose homeostasis, probably via boosting inflammatory state and consequent impairment of insulin metabolism [44,45]. It remains to be known whether the same changes would occur in pregnant women of normal weight who have GDM. Ma et al. carried out a case-control study and revealed that *Eisenbergiella*, *Tyzzerella 4*, and *Lachnospiraceae NK4A136* were enriched in the GDM group compared to controls who were matched with age, gestational age, and sample collection date [46]. They also discovered that *Eisenbergiella* and *Tyzzerella 4* were positively correlated with fasting blood glucose levels [46]. Functional annotation modules also showed that sphingolipid metabolism, starch, and sucrose metabolism were boosted, while lysine biosynthesis and nitrogen metabolism were decreased [46]. In a nested case-control study, it was demonstrated that Actinomyces, Adlercreutzia, Bifidobacterium, Coriobacteriaceae, *Lachnospiraceae* spp., and Rothia were considerably reduced, whereas Enterobacteriaceae, *Ruminococcaceae* spp., and Veillonellaceae were augmented in GDM women. Furthermore, Hu et al. affirmed the predictive power of microbial alterations; they demonstrated that the abundance of *Staphylococcus* relative to Clostridium, Coriobacteriaceae, and *Roseburia* was positively correlated with fasting blood glucose and postprandial glucose level [47].

Gut microbiota dysbiosis in women with GDM also occurs during the second trimester. Pathobionts, such as *P. distasonis*, *Klebsiella variicola*, and *Catenibacterium mitsuokai*, which were positively associated with maternal glucose levels, were revealed to be elevated during the second trimester in women with GDM compared to women without GDM. In comparison, the levels of beneficial butyrate-producing bacteria, including *Alistipess* spp., *Bifidobacterium* spp., *Eubacterium* spp., and *Methanobrevibacter smithii,* were lowered [17]. Moreover, functional analysis figured out that gut microbiota dysbiosis during the second trimester may be correlated with the upregulation of membrane transport, energy metabolism, lipopolysaccharide, and phosphotransferase activation, as well as the suppression of amino acid metabolic pathways [17]. Chen et al. confirmed that beneficial acetate-producing, lactate-producing, and butyrate-producing bacteria, including *Bifidobacterium* spp., *Eubacterium* spp., were depleted while *Corynebacterium* spp., *Lactobacillus* spp., and *Blautia hydrogenotrophica* were enriched in GDM patients [35]. Ye and Zhang reported a rise in the abundance of *Blautia* and *Eubacterium hallii group*, with a reduction in the richness of *Faecalibacterium* in GDM patients who failed to control glycemic [48]. It was affirmed that Gemmiger, Oscillospira, Enterococcaceae unassigned genera of Clostridiales, Ruminococcaceae, seven genera within the phylum Firmicutes, and two within the phylum Actinobacteria were depleted, while four genera within phylum Bacteroidetes were raised [36]. Simultaneously, the genera Bacteroides, Dialister, and Campylobacter, and an unassigned genus of Enterococcaceae appeared to be taxonomic biomarkers of GDM, whereas the genera Gemmiger and Bifidobacterium, as well as unassigned genera of Clostridiales and Ruminococcaceae, appeared to be markers of normal pregnancy [36]. Wei et al. carried out a cross-sectional study and discovered no difference in α-diversity, but they stated that *Ruminococcus bromii*, *Clostridium colinum*, and *Streptococcus infantis* were significantly associated with the GDM samples. Moreover, *S. infantis* were positively associated with blood glucose levels after adjusting for body mass index (BMI) [49]. A cross-sectional study in which BMI was matched in separate groups revealed no difference in β-diversity in women with GDM. However, the higher abundance of *Bacteroides caccae*, *Bacteroides massiliensis*, and *Bacteroides thetaiotaomicron*, with a lower abundance of *Bacteroides vulgatus*, *Eubacterium eligens*, *Lactobacillus rogosae*, and *Prevotella copri* were detected in women with GDM [42]. According to Zhang et al., women who later progressed to GDM showed decreased α-diversity and abundance of genera belonging to Ruminococcaceae, Coriobacteriales, and Lachnospiraceae [39]. Liang et al. explored that *Bacteroides* and *Lachnoclostridium* were more abundant, but *Ruminococcaceae UCG-002*, *Ruminococcaceae UCG-005*, *Clostridium sensu stricto 1*, and *Streptococcus* were reduced in the GDM group. Further, *Paraprevotella*, *Roseburia*, *Faecalibacterium*, and *Ruminococcaceae_UCG-002* were significantly negatively correlated with glucose. *Ruminococcaceae_UCG-002* was significantly negatively correlated with hemoglobin A1c. Bacteroides were significantly positively correlated with glucose, while *Sutterella*, *Oscillibacter*, and *Bifidobacterium* were significantly positively correlated with glucagon-like peptide-1 [50].

During the third trimester, the *Akkermansia* genus was enhanced in women with GDM [51], while the abundance of *Faecalibacterium* was found to decrease substantially [19]. According to Crusell et al., Actinobacteria at the phylum level and *Collinsella*, *Desulfovibrio*, and *Rothia* at the genus level were more abundant in GDM women [18]. After adjustment for BMI, OTUs allocated to *Akkermansia* were associated with decreased insulin sensitivity, while *Christensenella* OTUs were associated with greater fasting plasma glucose concentration in GDM women [18]. Crusell et al. also discovered that the aberrant composition of the gut microbiome could be observed till 8 months after birth [18]. According to Cortez et al., *Ruminococcus* and *Prevotella* were increased, while Bacteroides, *Eubacterium*, and Firmicutes were reduced in women with GDM during the third trimester [37,52]. In a cross-sectional study, increased Gammaproteobacteria and *Haemophilus* abundances were found in women with GDM compared to weight-matched controls [38]. However, Li and colleagues declared the greater α-diversity in women with GDM. They also affirmed greater BMI, a larger proportion of *Firmicutes*, and a lower proportion of *Bacteroides* than healthy controls [43]. Dynamic changes in intestinal flora from the first trimester to the second trimester were observed. It was also discovered that the GDM group had greater pregestational BMI, a consistent decrease in *Coprococcus* and *Streptococcus* levels, and suppressed interbacterial interactions, which may potentially serve as early biomarkers for GDM [53] (Table 1).

In a nutshell, despite many studies focusing on gut microbiota in GDM, the findings are inconclusive. Disparities in these findings can be attributed to sample size, geographical location, study design, and participant selection criteria (including population BMI, food habits, gestational age, and sequencing platform) [7]. Moreover, studies that dynamically investigate the changes through multi-point fecal sampling during different trimesters are rare. Still, it can be concluded that gut microbiota dysbiosis in GDM can be characterized by changes in α-diversity and β-diversity, an increase in gram-negative bacteria and part of gram-positive bacteria, alterations in SCFA-producing bacteria (including *Akkermansia*, *Bifidobacterium bifidum*, *Coprococcus*, *Lactobacillus casei*, *Roseburia*, and *Ruminococcus*), and a decrease in bacteria with probiotics properties [17,18,19,35,36,44,47,50,54].

## 4. GDM–Gut Interaction

### 4.1. Effect of Metabolic Overload and Chronic Inflammation on Gut Microecology

Currently, research into the underlying mechanisms involving GDM and gut microbiota dysbiosis is at the preliminary stage. Many studies have reported that changes in gut microecology were associated with decreased insulin sensitivity, higher plasma glucose concentration, and greater BMI in GDM women. These changes appear to begin in early pregnancy and last for several months after delivery, negatively affecting intestinal function.

Studies also indicated that pregnant women with GDM tend to have a significantly higher pre-pregnancy BMI than women without GDM [37,43]. Overweight and obesity before pregnancy are associated with a greater risk of developing GDM [55]. The main reason may be related to gut dysbiosis, which plays a role in the pathogenesis and progression of GDM [56]. Most studies revealed that in genetical obese ob/ob mice, diet-induced obese (DIO) mice, and obese people, the proportion of *Akkermancia muciniphila*, butyrate-producing bacteria, including *Roseburia* and *Faecalibacterium prauznitzii*, were reduced [57,58,59]. At the same time, the proportion of Bacteroidetes and Firmicutes was aberrant; though some studies agreed on a significant increase in Firmicutes, others reported a significant increase in Bacteroidetes and a decrease in Firmicutes [60,61,62,63,64,65,66]. Early white adipose tissue dilatation and chronic obesity can both activate the inflammatory programs, permanently skewing the immune system toward proinflammatory phenotypes [67].

GDM is usually accompanied by hyperglycemia and IR. Human metagenome-wide association studies in patients with type 2 diabetes demonstrated highly significant correlations between hyperglycemia and IR and specific intestinal bacteria, bacterial genes, and associated metabolic pathways [68]. Hyperglycemia affects goblet cell proliferation, maturation, and mucus biosynthesis, which will disrupt the intestinal barrier and intestinal mucus, allowing pathogenic bacteria and their elements, such as LPS, to pass through the epithelial barrier [69,70]. Subsequently, LPS binds to toll-like receptors on the surface of intestinal epithelial cells to recruit immune cells in the intestine [67,71]. IFN-γ, IL-1, and other proinflammatory cytokines are released by recruited immune cells, disrupting the enteral environment and increasing intestinal permeability, further eliciting an immune response and triggering chronic inflammation [70,72,73]. Hyperglycemia can also drive immune cell dysfunction via mitochondrial dysfunction [74,75]. Damage to the enteral environment and immune response will affect the value and growth of intestinal flora, leading to gut microbiota dysbiosis [67,71,76,77,78].

### 4.2. Roles of Gut Microecology in GDM

Hypotheses have been put forward that changes in microecology may be related to GDM by affecting intestinal barrier function, glucose metabolism disorder, energy accumulation, and other ways.

#### 4.2.1. Gut Microbiota Dysbiosis

Zonulin, which is released by the liver and gut epithelial cells, is a physiological tight junction modulator and a potential predictor of gastrointestinal permeability [7,79]. Recent research has found that the zonulin level in plasma is significantly elevated in GDM women, indicating their greater gastrointestinal permeability and dysfunction of the gut barrier [44,80]. The gut barrier consists of mucus, intercellular tight junctions, immunoglobulins, antimicrobial peptides secreted by Paneth cells, microbes, and other components [79]. The mucus barrier is one of the first lines of the gut barrier. Gut microbiota dysbiosis may result in the thinning of the mucosal layer in GDM patients [81]. For example, *Prevotella* and *Akkermansia* levels are reported to be elevated in women with GDM. They are mucin-degrading pathobionts, which can increase mucin oligosaccharide degradation beyond the normal limit, leading to gut barrier dysfunction and leaky gut [57,82]. Additionally, several studies have proved a decrease in SCFA-producing bacteria in women with GDM during the second and third trimesters [24,35]. Elhaseen et al. discovered that SCFAs could significantly upregulate the tight junction protein genes (ZO-1 and occludin) of Caco-2 cells, implying that SCFAs have the potential to strengthen the gut barrier [83]. Therefore, a decrease in SCFAs may be associated with increased permeability of the gut epithelium.

The other feature of gut microbiota dysbiosis in women with GDM is the elevated level of gram-negative bacteria, including *Prevotella*, *Haemophilus*, and *Desulfovibrio* in both mid and late pregnancy [18,37,38,42,52]. Rises in Gram-negative pathobionts are linked to LPS biosynthesis. Accordingly, functional analysis revealed that pathways relating to LPS biosynthesis and transport system were increased in women with GDM [17,52]. Elevated levels of Gram-negative bacteria and LPS in women with GDM may weaken the intestinal epithelial barrier [7]. Elevated LPS and pathobionts, on the other hand, can adhere to the mucosal layer, cross the epithelial layer of the gut via toll-like receptor 2/4 (TLR2/4) activation, translocate via phagocytosis and dendritic cell (DC) co-localization, ultimately entering systemic circulation and causing metabolic endotoxemia [7,79]. Furthermore, the compromised intestinal barrier can facilitate the migration of LPS and pathobionts, which enter the systemic circulation and peripheral tissues, binding to and activating TLR [70,72,73]. TLR activation triggers macrophage infiltration and inflammation pathways, such as C-Jun N-terminal kinase (JNK), inhibitory B kinase (IKK), and nuclear factor kappa-B (NF-κB), which might evoke serine phosphorylation of the insulin receptor substrate-1^+Ser307^ (IRS-1^+Ser307^), resulting in suppression of phosphatidylinositol 3-kinase (PI3-K) and protein kinase B^Ser473^ (Akt^Ser473^). This procedure will impair insulin signaling and reduce glucose uptake in peripheral tissues, resulting in hyperglycemia in GDM women [7,70,72,73] (Figure 1).

#### 4.2.2. Roles of Critical Metabolites

Gut-related metabolites include SCFAs, bile acids (BAs), amino acids, and so on. Metabolomics analysis has revealed that these metabolites were significantly altered in women with GDM, especially SCFAs.

SCFAs are derived from microbiota-accessible carbohydrates (MACs) fermented in the colon from dietary fibers and resistant starch fermentation [84,85]. They are extensively involved in processing undigested diets for additional energy, lipid metabolism, glucose metabolism, and inflammation, in addition to their potential role in gut barrier maintenance [52,83,86]. SCFAs promote adipogenesis in adipose tissue by increasing the expression of the peroxisome proliferator-activated receptor (PPAR) while decreasing serum FFA levels by suppressing lipolysis through activating GPR43 [87,88]. Propionate is an essential substrate for gluconeogenesis in the liver and skeletal muscle, which can also inhibit lipogenesis by suppressing fatty acid synthase [88]. Butyrate and acetate have been linked to lipogenesis in the liver [89]. SCFAs also endorse glycogen storage while inhibiting glycolysis in the liver and skeletal muscles [90]. In the gut, SCFAs can stimulate enteroendocrine L cells to release glucagon-like peptide-1 (GLP-1) and intestinal hormone peptide YY (PYY) [91,92,93]. They both are essential brain-gut peptides that help to control appetite, promote insulin secretion, and suppress the release of pancreatic glucagon [93,94]. On the other hand, butyrate constitutes a major energy source for intestinal epithelial cells and metabolic responses, while acetate regulates intestinal PH value and intestinal flora [54,95]. They are essential for enhancing epithelial barrier function and intestinal microecology. Moreover, SCFAs influence hematopoietic progenitors in the murine bone marrow and maintain the balance between anti-inflammatory and proinflammatory cells in mice by affecting peripheral DC cells and T cells, implying that they are important for the development of the innate and adaptive immune system [96,97].

As mentioned above, the amount and types of SCFAs are largely determined by the number of SCFA-producing bacteria. The Firmicutes phylum, particularly *Faecalibacterium*, *Roseburia*, and *Bifidobacterium*, produce butyrate, whereas *Bacteroidetes* produce acetate and propionate [98]. The majority of butyrate will be absorbed as an energy source by colonocytes (7), whereas the majority of acetate will be metabolized by the liver, adipose, muscle, heart, and kidney tissues [7,99].

Alterations in the abundance of SCFA-producing bacteria have been reported in women with GDM. The compositions of SCFA-producing genus *Faecalibacterium*, *Bifidobacterium*, *Ruminococcus*, *Roseburia*, *Coprococcus*, *Akkermansia*, *Phascolarctobacterium*, and *Eubacterium* were deficient in women with GDM and were inversely correlated with glucose tolerance [17,18,19,37,48,51,53]. As a result, acetate, butanoate, and propanoate production were revealed to be reduced during the second and third trimesters [95,100]. Inadequate SCFAs reduce adipose tissue lipid storage capacity, inhibit fatty acid oxidation, and increase lipolysis, which then raises serum FFA levels and increases lipid storage in the liver and muscle [7,48]. Simultaneously, insufficient SCFAs may be unable to maintain the balance of anti-inflammatory and proinflammatory cells, leading to low-grade inflammation. Elevated serum FFA levels, increased lipid storage in the liver and muscle, and low-grade inflammation may all contribute to insulin resistance and hyperglycemia in women with GDM [7,48]. At the same time, insufficient SCFAs may be unable to maintain the gut barrier and intestinal PH value, evoking gut leakage and gut dysbiosis and forming a vicious cycle [54]. On the other hand, elevations of SCFAs-producing bacteria such as Firmicutes, *Phascolarctobacterium*, *Faecalibacterium*, and *Bacteroidetes* also occur in women with GDM, which lead to excessive butyrate, isobutyrate, isovalerate, acetic and propionic production [52,101]. Excessive SCFAs production may lead to extra energy harvesting capacity and FFA overflow, increasing lipid storage in the liver and skeletal muscle and leading to obesity [102]. Additionally, it may upregulate gluconeogenesis pathways and suppress glycolysis pathways and insulin signaling in peripheral tissues, resulting in hyperglycemia and IR [7,102]. Both isobutyric and isovaleric acids are branched-chain SCFAs, which may be involved in the release of proinflammatory cytokines [101] (Figure 2).

Apart from the effect of SCFA on energy storage, changes in the gut microbiota of women with GDM may also have a direct impact on energy storage. It was reported that bacteria associated with energy storage promotion, such as *Akkermansia*, *Bifidobacterium*, and Firmicutes, were augmented during the third trimester in healthy pregnant women. However, bacterium levels appeared to be lower in the second trimester [35,36,47], whereas *Akkermansia* levels were higher in the third trimester in women with GDM [18,51]. Hence, women with GDM may experience energy storage disorders, eliciting energy metabolism disturbance, which may be associated with glucose and lipid metabolism disorders [58,62,86]. On the other hand, *Bacteroidetes* was reported to be elevated in GDM women and was positively correlated with blood glucose. *Bacteroidetes* is considered a gram-negative bacterium that can produce a proinflammatory marker—LPS [75]. It may be implicated in the pathogenesis and progression of metabolic disturbance via inflammation [17,38,103].

Moreover, aromatic amino acids (AAA)-degrading bacteria such as *Clostridium*, *Fusobacterium*, and *Eubacterium* were decreased in GDM women compared to healthy pregnant women [17,37]. That might be the reason for the notably delayed and blunted decrease in AAA in GDM women [104]. Since AAA catabolism by the gut microbiome yields numerous bioactive molecules that regulate murine immune, metabolic, and neuronal responses in the gut and distant organs, decreased AAA catabolism may disturb host–microbe metabolic axes [105]. This needs to be further clarified in human trials. Metabolomics analysis of meconium and serum showed that metabolic pathways, including taurine and hypotaurine metabolism, pyrimidine metabolism, beta-alanine metabolism, and BAs biosynthesis, were altered in GDM subjects [106]. However, the correlation between these changes in metabolites and intestinal flora and GDM still remains unclear.

## 5. Prospect and Implication

As GDM has become a major health burden for both mothers and children, it is critical to propose effective prevention and treatment strategies. Nonetheless, pregnancy is a unique time, and treatment must take the safety of the fetus into account. In the United States, insulin is currently the first-line agent recommended for the treatment of GDM, whereas metformin and glyburide are not because they can cross the placenta to the fetus and cause neonatal hypoglycemia [107]. Hence, the current options for pharmacologic therapy are extremely limited. Given the importance of gut microecology in GDM, regulating intestinal flora may be a potential strategy for GDM (Figure 3).

Lifestyle behavior change, including dietary restriction and exercise, is the first line of GDM management [107], which are well-established to have the greatest impact on gut microbiota [108]. Dietary carbohydrates (CHO) are an important source of energy for both the mother and the fetus, which can impact gut microbiota and blood glucose levels. Nondigestible CHO components (fiber) can serve as an energy source for colonic microbiota, modulating gut microbiota composition [109]. Lower fiber intake has been linked to decreased gut microbiota diversity and richness [110], increased abundance of *Collinsella* (a genus closely linked to higher blood glucose) and *Sutterella* (a Proteobacteria linked to inflammatory status), and higher serum zonulin levels [111,112,113,114]. Furthermore, monounsaturated fatty acids (MUFAs) have been associated with an increase in the abundance of Firmicutes, Proteobacteria, and Bacteroidetes [113]. In pregnant women, fat-soluble vitamins also appear to modulate gut microbiota. Mandal et al. conducted the cohort study and discovered that higher vitamin D intake was associated with decreased microbial α-diversity and a relative decrease in the abundance of proinflammatory Proteobacteria phylum [113]. Higher vitamin E intake was also coupled with a decrease in the abundance of Proteobacteria [113]. To control maternal fasting and postprandial glucose during GDM, current dietary guidelines recommend limiting CHO intake or replacing high glycemic/rapidly digesting CHO with those that are more slowly digesting, as well as taking enough vitamins [107].

Exercise, on the other hand, has been shown to strengthen gut health by increasing microbiome diversity and balancing beneficial and pathogenic bacterial communities [115,116,117]. Specifically, exercise increases butyrate-producing bacteria such as *Roseburia hominis*, which raises butyrate concentrations in both humans and mice [116,118]. It can also boost key antioxidant enzymes and anti-inflammatory cytokines in intestinal lymphocytes, assisting in the reduction of intestinal inflammation [116,118].

Probiotics and prebiotics are live microorganisms that benefit the host when administered in adequate amounts [119]. As previously stated, women with GDM had a decrease in bacteria with probiotic and prebiotic properties. Therefore, probiotic and prebiotic supplementation may be advantageous. Parallel, double-blind, randomized control trials (RCTs) were published by Dolatkhah et al. They enrolled women with GDM at 24 to 30 weeks of gestation and randomly assigned them to receive probiotics or placebo in capsule form for 6 to 8 weeks. Participants who received probiotics experienced a decrease in fasting blood sugar (FBS) and a significant decrease in the homeostatic model assessment of IR (HOMA-IR) after treatment, implying the potential roles of probiotics on glucose regulation [120,121]. In another RCT, women with GDM were randomly assigned to receive either Probiotic Mixture (VSL#3 probiotics) in a capsule or a placebo for 8 weeks. Hs-CRP, TNF-α, and IL-6 levels were significantly lower in participants who received probiotics, disclosing the potential effect of probiotics on the immune response [122]. However, no effect of probiotics on glycemic control was found in women with GDM receiving a single-strain probiotic capsule delivering 1 billion CFU of *Lactobacillus salivarius UCC118* [123]. Differences between studies could be attributed to differences in sample size, probiotic dosage, and duration of probiotic/fermented food/antibiotic consumption prior to intervention [123,124]. Therefore, more RCTs are needed to determine which probiotic or prebiotic strains are optimal and how to utilize them for treating GDM. On the other hand, consuming probiotics and prebiotics requires constant monitoring to avoid adverse effects such as systemic infections and mild gastrointestinal upset [125].

The scope of this review is limited because research into the gut and GDM is still in its early stages. Moreover, the results may be affected by the different study designs and participant selection criteria (diet type, geographical environment, lifestyle, gestational age, and drug factors). We believe it is necessary to further investigate the changes in intestinal flora before the occurrence of hyperglycemia in GDM to provide new approaches for GDM prevention.

## 6. Conclusions

Although the abnormalities of the intestinal tract may not be the initial factor of GDM, it has been considered to participate in the pathological process of GDM greatly. Women with GDM have significantly distinct gut microbiota and gut metabolites than healthy pregnant women. In turn, the alerted gut microecology and related metabolites substantially impact intestinal barrier function, inflammation, insulin signaling, glucose metabolism, lipid metabolism, and energy accumulation, playing a significant role in both the pathogenesis and progression of GDM. Dietary restrictions, exercise, and probiotics (prebiotics), which can reverse the altered gut microbiota and metabolites, may help alleviate symptoms of GDM.

## Figures and Tables

**Figure 1 microorganisms-11-01725-f001:**
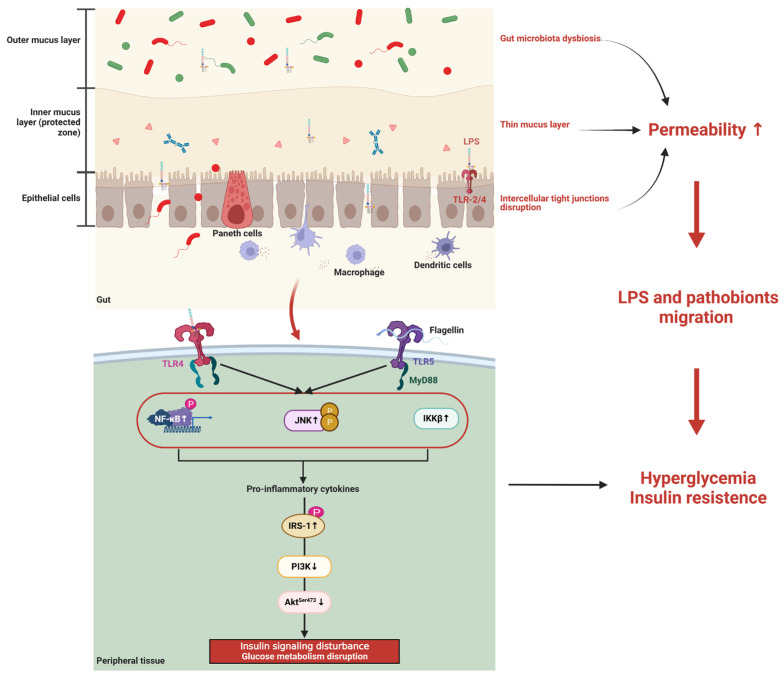
Roles of gut microbiota dysbiosis in GDM. Figure legend: Akt^Ser473^: Protein kinase B^Serine473^; IKK: Inhibitory B kinase; IRS1: Insulin receptor substrate-1; JNK: C-Jun N-terminal kinase; LPS: Lipopolysaccharide; NF-κB: Nuclear factor kappa-B; PI3-K: Phosphatidylinositol 3-kinase; TLR: Toll-like receptor. The gut barrier consists of mucus, intercellular tight junctions, immunoglobulins, antimicrobial peptides secreted by Paneth cells, microbes, and other components. Raises in Gram-negative pathobionts are linked to LPS biosynthesis. Elevated LPS and pathobionts can adhere to the mucosal layer, cross the epithelial layer of the gut via TLR2/4 activation, and translocate via phagocytosis and DC co-localization, ultimately entering systemic circulation and causing metabolic endotoxemia. Further, gut microbiota dysbiosis may result in thinning of the mucosal layer and disruption of the intercellular tight junctions in GDM patients. The compromised intestinal barrier can facilitate the migration of LPS and pathobionts. LPS and pathobionts bind to and activate TLR. TLR activation triggers macrophage infiltration and inflammation pathways, such as JNK, IKK, and NF-κB, which might evoke serine phosphorylation of IRS-1+Ser307, resulting in suppression of PI3-K and AktSer473. This procedure will impair insulin signaling and reduce glucose uptake in peripheral tissues, resulting in hyperglycemia in GDM women. The figure was created with BioRender.com. URL: https://biorender.com (accessed on 24 June 2023).

**Figure 2 microorganisms-11-01725-f002:**
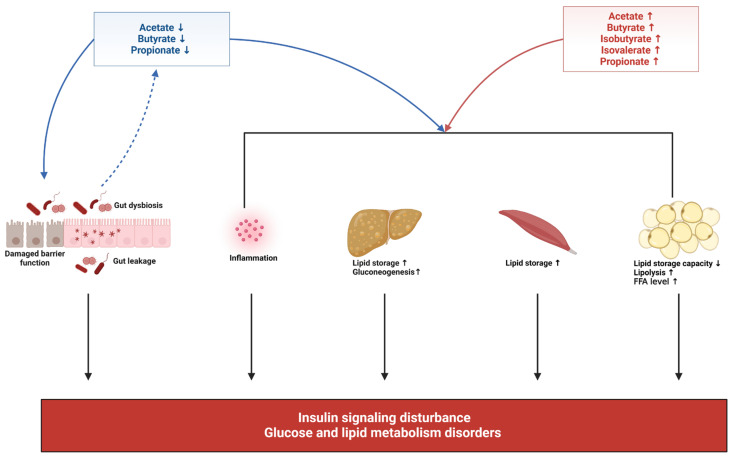
Roles of SCFAs in GDM. Figure legend: Inadequate SCFAs may be associated with increased permeability of gut epithelium. Inadequate SCFAs also reduce adipose tissue lipid storage capacity, inhibit fatty acid oxidation, and increase lipolysis, which then raises serum FFA levels and increases lipid storage in the liver and muscle. Simultaneously, insufficient SCFAs may be unable to maintain the balance of anti-inflammatory and proinflammatory cells, leading to low-grade inflammation. Elevated serum FFA levels, increased lipid storage in the liver and muscle, and low-grade inflammation may all contribute to insulin resistance and hyperglycemia in women with GDM. On the other hand, excessive SCFA production may lead to extra energy harvesting capacity and FFA overflow, increasing lipid storage in the liver and skeletal muscle and leading to obesity. Excessive SCFAs may upregulate gluconeogenesis pathways and suppress glycolysis pathways and insulin signaling in peripheral tissues, resulting in hyperglycemia and insulin resistance. The figure was created with BioRender.com. URL: https://biorender.com (accessed on 24 June 2023).

**Figure 3 microorganisms-11-01725-f003:**
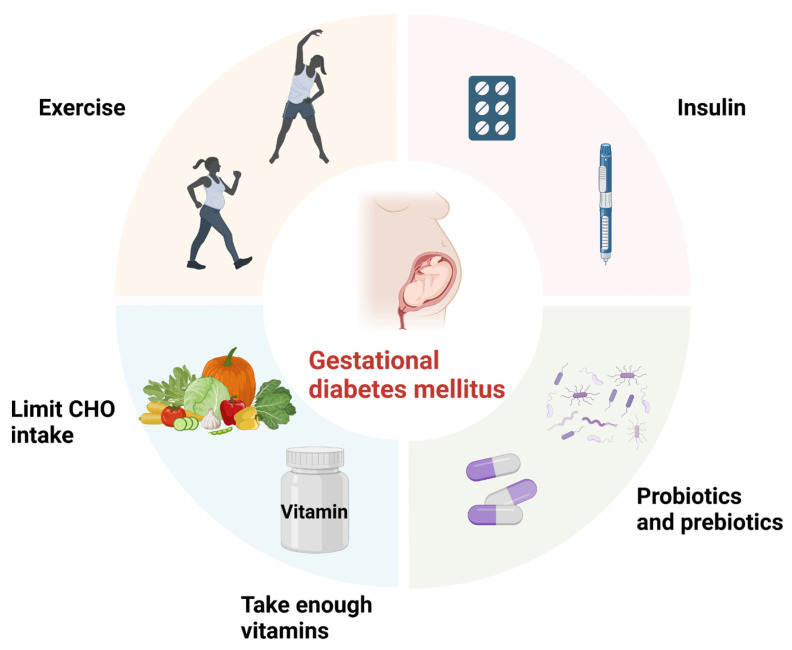
Treatment for GDM. Figure legend: CHO: Carbohydrates. Dietary restrictions, exercise, probiotics (prebiotics), and insulin may help alleviate symptoms of GDM. The figure was created with BioRender.com. URL: https://biorender.com (accessed on 24 June 2023).

**Table 1 microorganisms-11-01725-t001:** Characteristics of included studies of gut microbiota and related metabolic changes during GDM.

Author, Year	Study Design	Group Setting	Time	Measurements	Conclusions
During the first trimester					
Mokkala, 2017 [44]	Case-control study	15 Women who developed GDM VS. 60 healthy pregnant women	10.4–15.4 weeks of gestation	16S RNA gene sequencing, QIIME pipeline	1. Ruminococcaceae family were increased in GDM women; 2. Ruminococcaceae family ≥26.8% predicted positive GDM diagnosis (sensitivity: 60%, specificity: 88%).
Ma, 2020 [46]	Case-control study	98 Women who developed GDM VS. 98 healthy pregnant women	10–15 weeks of gestation	16S rRNA microarray, Shotgun metagenomics sequencing	1. *Eisenbergiella*, *Tyzzerella 4*, and *Lachnospiraceae NK4A136* were enriched in the GDM group; 2. *Eisenbergiella* and *Tyzzerella 4* were positively correlated with FBS.
Hu, 2021 [47]	Case-control study	201 Women who developed GDM VS. 201 matched controls	6–15 weeks of gestation	16S rRNA microarray	1. Actinomyces, Adlercreutzia, Bifidobacterium, Coriobacteriaceae, *Lachnospiraceae* spp., and Rothia were reduced in GDM women; 2. Enterobacteriaceae, *Ruminococcaceae* spp., and Veillonellaceae were augmented in GDM women 3. Abundances of Staphylococcus relative to Clostridium, Coriobacteriaceae, and Roseburia were positively correlated with FBS and postprandial glucose levels.
During the second trimester					
Kuang, 2017 [17]	Case-control study	43 GDM patients VS. 81 healthy pregnant women	21–29 weeks of gestation	Whole-metagenome shotgun sequencing	1. *P. distasonis*, *Klebsiella variicola*, and *Catenibacterium mitsuokai* were elevated in women with GDM, which were positively associated with maternal glucose levels; 2. *Alistipess* spp., *Bifidobacterium* spp., *Eubacterium* spp., and *Methanobrevibacter smithii* were lowered.
Chen, 2021 [35]	Case-control study	30 GDM patients VS. 28 healthy pregnant women	24–28 weeks of gestation	16S rRNA microarray	1. *Bifidobacterium* spp., and *Eubacterium* spp., were depleted in GDM patients; 2. *Corynebacterium* spp., *Lactobacillus* spp., and *Blautia hydrogenotrophica* were enriched in GDM patients.
Ye, 2019 [48]	Case-control study	24 GDM patients with successful glycemic control vs. 12 failure of glycemic control	24–28 weeks of gestation	16S rRNA sequencing	1. The abundance of *Blautia* and *Eubacterium hallii group* was augmented in GDM patients who failed to control glycemic; 2. The richness of *Faecalibacterium* was reduced in GDM patients who failed to control glycemic.
Chen, 2021 [36]	Case-control study	110 GDM patients VS. 220 healthy pregnant women	22–24 weeks of gestation	16S rRNA sequencing	1. GDM patients had lower α-diversity that was significantly associated with glycemic; 2. Seven genera within the phylum Firmicutes and two within the phylum Actinobacteria were decreased in GDM patients; 3. Four genera within phylum Bacteroidetes were elevated in GDM patients.
Wei, 2022 [49]	Cross-sectional study	15 GDM patients VS. 18 healthy pregnant women	24–28 weeks of gestation	16S rRNA gene amplicon sequencing	1. No difference in α-diversity; 2. *Ruminococcus bromii*, *Clostridium colinum*, and *Streptococcus infantis* were increased in GDM patients; 3. *S. infantis* were positively associated with blood glucose levels after adjusting for BMI.
Festa, 2020 [42]	Cross-sectional study	14 GDM patients VS. 15 healthy pregnant women matched on BMI	24–28 weeks of gestation	16S rRNA microarray	1. No difference in β-diversity in women with GDM; 2. Abundances of *Bacteroides caccae*, *Bacteroides massiliensis*, and *Bacteroides thetaiotaomicron* were elevated in GDM patients; 3. Abundance of *Bacteroides vulgatus*, *Eubacterium eligens*, *Lactobacillus rogosae*, and *Prevotella copri* were suppressed in women with GDM.
Zhang, 2021 [39]	Prospective cohort study	128 GDM patients VS. 709 healthy pregnant women	22–24 weeks of gestation	16S rRNA gene sequencing	1. Women who later progressed to GDM showed decreased α-diversity; 2. Genera belonging to Ruminococcaceae, Coriobacteriales, and Lachnospiraceae were decreased in GDM patients
Liang, 2022 [50]	Case-control study	35 GDM patients VS. 25 healthy pregnant women	24–28 weeks of gestation	16S rRNA gene sequencing	1. Abundances of Bacteroides and Lachnoclostridium were elevated in GDM patients; 2. *Ruminococcaceae UCG-002*, *Ruminococcaceae UCG-005*, *Clostridium sensu stricto 1*, and *Streptococcus* were reduced in GDM patients; 3. *Paraprevotella*, *Roseburia*, *Faecalibacterium*, and *Ruminococcaceae_UCG-002* were negatively correlated with glucose; 4. *Ruminococcaceae_UCG-002* was negatively correlated with hemoglobin A1c; 5. *Bacteroides* was positively correlated with glucose; 6. *Sutterella*, *Oscillibacter*, and *Bifidobacterium* were positively correlated with GLP-1.
During the third trimester					
Wang, 2018 [19]	Cross-sectional study	581 maternal and 248 neonatal samples	During the third trimester	16S rRNA gene sequencing	The abundance of *Faecalibacterium* was reduced in GDM patients.
Crusel, 2018 [18]	Case-control study	50 GDM patients VS. 157 healthy pregnant women	During the third trimester and 8 months postpartum	16S rRNA gene amplicon sequencing	1. Actinobacteria at the phylum level and *Collinsella*, *Desulfovibrio*, and *Rothia* at the genus level were enriched in GDM women; 2. After adjustment for BMI, OTUs allocated to *Akkermansia* were associated with decreased insulin sensitivity, while *Christensenella* OTUs were associated with greater fasting plasma glucose concentration in GDM women; 3. Aberrant composition of the gut microbiome can be observed till 8 months after birth.
Cortez, 2019 [37]	Cross-sectional study	26 GDM patients VS. 42 healthy pregnant women	28–36 weeks of gestation	Next-generation sequencing	1. Abundances of Firmicutes phylum were increased in the GDM group, while that of the Bacteroidetes phylum was increased in the control group; 2. GDM patients have a higher ratio of Firmicutes/Bacteroidetes.
Ferrocino, 2018 [52]	Cohort study	41 GDM patients	24–28 weeks and at 38 weeks of gestation	16S amplicon-based sequencing	Firmicutes were enriched, and Bacteroidetes and Actinobacteria were decreased in GDM patients.
Xu, 2020 [38]	Cross-sectional study	30 GDM patients VS. 31 healthy pregnant women	During the third trimester	16S rRNA sequencing	1. GDM cases showed lower α-diversity; 2. Abundances of Gammaproteobacteria and Haemophilus were augmented in women with GDM
Li, 2021 [43]	Cross-sectional study	23 GDM patients VS. 29 healthy pregnant women	>28 weeks of gestation	16S rRNA sequencing	1. Greater α-diversity in women with GDM; 2. A larger proportion of Firmicutes and a lower proportion of Bacteroides in women with GDM.
From the first trimester to the second trimester					
Zheng, 2020 [53]	Cross-sectional study	31 GDM patients VS. 103 healthy pregnant women	During the first half of pregnancy	16S rRNA gene amplicon sequencing	*Coprococcus* and *Streptococcus* levels were consistently reduced in women with GDM.

BMI: body mass index, FBS: fasting blood glucose, rRNA: ribosomal ribonucleic acid.

## Data Availability

Not applicable.

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
