# Peer review of "Gut Microbiota and Critical Metabolites: Potential Target in Preventing Gestational Diabetes Mellitus?"

_microorganisms, 2023, doi:10.3390/microorganisms11071725_

Round 1

Reviewer 1 Report

The review by Hu and colleagues highlights potential metabolomic targets in women affected from gestational diabetes mellitus.

Minor modifications are required.

Table 1 needs a restyling. The three trimesters deserve at least to be framed in a more clearer way. 

Please explain the meaning of "microecology" related to GDM. Moreover, the written concept is too short and is not sufficient alone to support the subparagraph.

This reviewer thinks that it would be useful to create a figure resuming the most important metabolic pathways involved.

Author Response

Comments and Suggestions for Authors

Reviewer 1: The review by Hu and colleagues highlights potential metabolomics targets in women affected from gestational diabetes mellitus.

Minor modifications are required.

  1. Table 1 needs a restyling. The three trimesters deserve at least to be framed in a clearer way.

Answer: Thanks to the reviewers' suggestions, we have revised the Table 1 (see Table 1 for details).

  1. Please explain the meaning of "microecology" related to GDM. Moreover, the written concept is too short and is not sufficient alone to support the subparagraph.

Answer: We are sorry for any confusion and thanks to the reviewers' suggestions. We have revised the part. In line 68, we introduced that gut microecology meant the great diversity of microorganisms and its surrounding environment. Gut microbiota was the most important component of gut microecology, although intestinal mucosa had a significant impact on gut microecology. Moreover, the changes in gut microbiota and metabolites in GDM women was showed in section 2 “Gut microbiota and related metabolic changes during GDM”. Further, we proposed that these changes may be related to pathogenesis and progression of GDM by affecting intestinal barrier function, glucose metabolism disorder, and energy accumulation, forming a vicious loop.

The changed microecology related to GDM referred to the altered gut microbiota and related metabolic changes introduced in section 2 (see the modified part of the text for details).

  1. This reviewer thinks that it would be useful to create a figure resuming the most important metabolic pathways involved.

Answer: Thanks to the reviewers' suggestions, we have added the figures of the mechanisms, interactions, and potential implications (see Figure 1-3 for details).

Reviewer 2 Report

It is quite an interesting paper. The purpose of this review is to summarize the impacts and mechanisms of gut microbiota and related metabolites in gestational diabetes mellitus (GDM), as well as to suggest prospective intervention targets, thus providing potential intervention targets. The Authors preciously described: gut microbiota changes during normal pregnancy, gut microbiota and related metabolic changes during GDM, GDM-gut interaction, prospect, and implication.

The Authors should address the following issues:

1.      Please add the conclusion section

2.      It will be interesting to add some figures, which could allow the readers to better understand the mechanisms, interactions, and possibles implications

3.      The references did not include any authors’ names

Author Response

Reviewer 2: It is quite an interesting paper. The purpose of this review is to summarize the impacts and mechanisms of gut microbiota and related metabolites in gestational diabetes mellitus (GDM), as well as to suggest prospective intervention targets, thus providing potential intervention targets. The Authors preciously described: gut microbiota changes during normal pregnancy, gut microbiota and related metabolic changes during GDM, GDM-gut interaction, prospect, and implication.

The Authors should address the following issues:

  1. Please add the conclusion section

Answer: Thanks to the reviewers' suggestions, we have added the conclusion section: “Although the abnormalities……may help alleviate symptoms of GDM” (see the modified part of the text for details).

  1. It will be interesting to add some figures, which could allow the readers to better understand the mechanisms, interactions, and possible implications

Answer: Thanks to the reviewers' suggestions, we have added the figures of the mechanisms, interactions, and potential implications (see Figure 1-3 for details).

  1. The references did not include any authors’ names

Answer: We are sorry for the mistake and thanks to the reviewers' suggestions. We have updated the references and have included the names of the authors.

Round 2

Reviewer 2 Report

The Authors included all my remarks and in my opinion the paper my be published in the current form.